:ưPLOS | ONE

# Resistance associated metabolite profiling of Aspergillus leaf spot in cotton through non-targeted metabolomics

**Maria Khizar[1], Jianxin Shi[2], Sadia Saleem[1], Fiza Liaquat[3], Muhammad Ashraf[2], Sadia Latif[1], Urooj Haroon[1], Syed Waqas Hassan[4], Shafiq ur Rehman[5], Hassan Javed Chaudhary[1], Umar Masood Quraishi[1], Muhammad Farooq Hussain Munis[1]** *

**1** Department of Plant Sciences, Faculty of Biological Sciences, Quaid-i-Azam University, Islamabad, Pakistan, **2** School of Life Sciences and Biotechnology, Shanghai Jiao Tong University, Shanghai, China, **3** School of Agriculture and Biology, Shanghai Jiao Tong University, Shanghai, China, **4** Department of Bioscience, University of Wah, Quaid Avenue, Wah Cantt., Pakistan, **5** College of Earth and Environmental Sciences, University of the Punjab, Lahore, Pakistan

* munis@qau.edu.pk

**Data Availability Statement:** All relevant data are within the manuscript and its Supporting Information files.

## Abstract

*Aspergillus tubingensis* is an important pathogen of economically important crops. Different biotic stresses strongly influence the balance of metabolites in plants. The aim of this study was to understand the function and response of resistance associated metabolites which, in turn are involved in many secondary metabolomics pathways to influence defense mechanism of cotton plant. Analysis of non-targeted metabolomics using ultra high performance liquid chromatography–mass spectrometry (UPLC-MS) revealed abundant accumulation of key metabolites including flavonoids, phenylpropanoids, terpenoids, fatty acids and carbohydrates, in response to leaf spot of cotton. The principal component analysis (PCA), orthogonal partial least squares discriminant analysis (OPLS-DA) and partial least squares discriminant analysis (PLS-DA) score plots illustrated the evidences of variation between two varieties of cotton under mock and pathogen inoculated treatments. Primary metabolism was affected by the up regulation of pyruvate and malate and by the accumulation of carbohydrates like cellobiose and inulobiose. Among 241 resistance related (RR) metabolites, 18 were identified as resistance related constitutive (RRC) and 223 as resistance related induced (RRI) metabolites. Several RRI metabolites, identified in the present study were the precursors for many secondary metabolic pathways. These included phenylpropanoids (stilbenes and furanocoumarin), flavonoids (phlorizin and kaempferol), alkaloids (indolizine and acetylcorynoline) and terpenoids (azelaic acid and oleanolic acid). Our results demonstrated that secondary metabolism, primary metabolism and energy metabolism were more active in resistant cultivar, as compared to sensitive cultivar. Differential protein and fatty acid metabolism was also depicted in both cultivars. Accumulation of these defense related metabolites in resistant cotton cultivar and their suppression in susceptible cotton cultivar revealed the reason of their respective tolerance and susceptibility against *A. tubingensis*.

**Funding:** This work was supported by: Maria Khizar, IRSIP-39-BMS-41, Higher education commission, Pakistan, https://hec.gov.pk/english/scholarshipsgrants/IRSIP/Pages/default.aspx. The funders had no role in study design, data collection and analysis, decision to publish, or preparation of the manuscript.

**Competing interests:** The authors have declared that no competing interests exist.

## Introduction

Cotton (*Gossypium hirsutum* L.) is a major fiber and cash crop and it is termed as "silver fiber" in different parts of the world. Pakistan ranks 4th among all cotton producing countries and its production plays key role in the economy of Pakistan [1]. There are about 50 species of cotton and four of them are cultivated, worldwide [2]. Cotton is a shrub and it is native to tropical and subtropical regions around the world, including America and Africa [3]. Eighty-seven percent of total cotton is grown in developing countries. In addition to fiber, cotton seed offers a supplemental income and it is a source of protein for human and animal nutrition. *G. hirsutum* is the most commonly grown cotton species and it is native to Mexico and Central America [4]. Cross breeding between diverse upland varieties helped the introduction of upland cotton in different areas of the world [5]. *G. hirsutum* is a natural allotetraploid species, that possibly rose from interspecific hybridization between ancestral diploid species, having an A-like genome (present day *G. arboreum*) and a D-like genome (present day *G. raimondii*) [6]. The upland form of *G. hirsutum* and its derived varieties are the backbone of the worldwide textile industry [7].

Cotton crop is affected by several biotic and abiotic stresses. In Pakistan, insect attack is the main cause of low crop yield [8]. Different fungal, bacterial, viral and pest diseases also affect the productivity of cotton plant, severely. Key fungal diseases of cotton include anthracnose caused by *Colletotrichum*, Ascochyta blight caused by *Ascochyta gossypii*, Charcoal rot caused by *Macrophomina phaseolina*, Fusarium wilt caused by *Fusarium oxysporum*, and leaf spot caused by *Altenaria* spp, and *Rhizoctonia solani* [3]. In recent years, leaf spot of cotton has also been reported to be caused by *Corynespora cassiicola* [9] and *Curvularia verruculosa* [10]. Among different *Aspergillus* species, *A. tubingensis* is a black filamentous species [11]. Mostly, *A. tubingensis* strain is misidentified as *A. niger* [12]. *A. tubingensis* has been reported to cause leaf spot disease of *Jatropha curcas* [13]. *A. tubingensis* is also a causal agent of bunch rot of shine muscat grape [14].

Metabolomics is a post-genomics tool to reveal physiological and biochemical responses of host under biotic and abiotic stresses [15]. Non-targeted metabolomics has been applied to interpret the host biochemical mechanism of quantitative resistance in crop plants against many pathogens [16]. Metabolomics profiling helps scientists to draw useful conclusions about the defense mechanism of commercially important crops. In wheat, a non-targeted metabolic profiling of wheat rachis revealed thickening of cell walls, due to the deposition of hydroxycinnamic acid amides (HCAAs) [17]. Accumulation of numerous RR metabolites has been reported as the reason of the growth inhibition of *F. graminearum* [18]. Non-targeted metabolic profiling has also been documented in Solanaceous crops including tobacco and potato. Host defensive phenylpropanoids (HCAAs) and fatty acids were induced in tobacco leaves under compatible interaction with *Phytophthora parasitica* [19]. Metabolic profiling of potato leaves infected with *Phytopthora infestans* revealed the induction of phenylalanine, tyrosine, shikimic acid, and benzoic acid, which are precursor metabolites for many defense related secondary metabolites [20]. Metabolomics profiling of cotton reveals that different species of genus Gossypium have different quantities of glycosides, such as, rhamnoglucosides are more abundant in *G. hirsutum* but are found in trace amounts in *G. barbadense* whereas kaempferol-3-glucoside and quercetin-7-glycosides are widely present in *G. barbadense*, in comparison to *G. hirsutum* [21]. Quercitin, kaempferol, bhenic acid, quercetin-3-rhamnoglucoside, catechin, epicatechin, scopoletin, gallocatechin, cinnamic acid, gossypol and stigmasterol are the known naturally occurring secondary metabolites of *G. hirsutum* [4]. The leaves of *G. hirsutum* contain 19 flavonoids spanning five different classes however there are no reports of flavones and chalcones or aurones from *G. hirsutum*. Higher concentrations of catechin, gallocatechin and isoquercitrin, found in young cotton leaves, result in halted mycelia growth [22].

The objectives of current study were to compare variation in metabolomics profiling of two cotton varieties which were subjected to Aspergillus leaf spot and to identify RR metabolites which were responsible for imparting resistance in tolerant variety.

## Materials and methods

### Plant material and growth conditions

Cotton seeds of susceptible cultivar (CIM-573) and resistant cultivar (NIA-Sadori) were obtained from Central Cotton Research Institute (CCRI), Multan and Nuclear Institute of Agricultural Science, Tandojaam, Pakistan, respectively. CIM-573 has been reported susceptible to bacterial leaf blight of cotton [23], while NIA-Sadori is known to exhibit resistance against biotic stresses [24].

Soil was prepared by mixing equal proportions of peat moss and clay (1:1 ratio). Healthy cotton seeds were surface sterilized using 2% sodium hypochlorite solution for 3 min and washed with distilled water, twice. These seeds were soaked overnight in double distilled water to increase germination potential of seeds [25]. Seeds were sown in 12 inch pots (4 seeds per pot) under greenhouse conditions at 32 ˚C, 70% relative humidity and 16/8 h of light/dark photoperiod for 5 weeks. Plants were fertilized fortnightly with 150 mL solution containing 20–20–20 NPK trace elements.

### Inoculum preparation and point-inoculation

*A. tubingensis* was grown on potato dextrose agar (PDA) media at 28 ˚C, under dark conditions. After 5 days of incubation, front and back sides of inoculated petri plates were carefully observed to see the morphology of mycelia. Microscopic study of growing mycelia was also performed. For the production of virulent spores, pathogen was inoculated on surface-sterilized leaves and placed on agar nutrient media for 3–4 days, at 25 ˚C. Fungal mycelium, grown on media plates were scraped using cultural loop to harvest sporangia and grown in czapek broth medium. Spores were filtered through double layer of cheesecloth and the spore concentration was adjusted to $1 \times 10^5$ mL$^{-1}$, using haemocytometer. Fully grown leaves of 5 to 7 week-old cotton plants were point inoculated with 20 μL of spore suspension (designated as inoculated plants) and sterile water (designated as mock plants). Four days post inoculation (dpi), leaves of both varieties along with their biological replicates containing inoculation site were cut, using a pair of sterile scissors, immediately frozen in liquid nitrogen and stored at −80 ˚C, until further use.

### Experimental design

The experiment was conducted in a randomized complete block (RCB) design. The experiment was consisting of four treatments: RT (resistant variety treated with pathogen), RM (resistant variety with mock treatment), ST (susceptible variety treated with pathogen), and SM (susceptible variety with mock treatment). Each treatment consisted of four biological replicates and the entire experiment was repeated three times, over a time interval of 3–4 days.

### Disease severity assessment

The experimental units were consisting of 12 leaves from at least three plants per replicate. The necrotic lesion diameter was measured using graph method at 4 dpi to calculate area under the disease progress curve (AUDPC).

## Liquid chromatography-mass spectrometry (UPLC-Q TOF-MS)

For this purpose, 80 mg of each leaf sample was weighed and transferred to Eppendorf tube. As internal standard, 20 μL of 2-chloro-l-phenylalanine (0.3 mg/mL methanol) and 1 mL mixture of methanol and water (7:3 v/v) were added to each sample and placed at -80 ˚C, for 2 min. Two small steel balls (pre-cooled at –20 ˚C, for 2 min) were added and grinded at 60 Hz for 2 min. The material was ultra-sonicated for 30 min and allowed to stand at –20 ˚C for 20 min. At 4 ˚C, these samples were centrifuged for 10 min at 13000 rpm. The supernatant (200 μL) was pipetted out and filtered using 0.22 μm organic phase pinhole filter. Samples were transferred to LC injection vials and stored at –80 ˚C, until LC-MS analysis. Quality control (QC) samples were prepared by mixing equal aliquots of all samples and each QC volume was the same as the sample. All extraction reagents were pre-cooled at -20 ˚C before use.

Liquid chromatography system consisting of Waters ACQUITY UPLC I-Class system (Waters Corporation, Milford, USA) coupled with VION IMS QTOF Mass spectrometer (Waters Corporation, Milford, USA) was used to analyze the metabolic profiling in both ESI positive and ESI negative ion modes. Chromatographic conditions included a column ACQUITY UPLC BEH C18 of 100 mm × 2.1 mm × 1.7 um. Column temperature was maintained at 45 ˚C. The mobile phase was comprised of both water and acetonitrile, containing 0.1% formic acid. Samples were kept at 4 ˚C during analysis. Flow rate was maintained at 0.4 mL min$^{-1}$ and injection volume was set at 2 μL. Mass spectrometry conditions included an ion source. Signal acquisition for mass spectrometry was accomplished using positive and negative ion scanning mode. Resolution of mass was set at 50–1000 amu with a scan time of 0.1 s and scan type was MS$^E$ in centroid mode. Data acquisition was performed in full scan mode (m/z ranges from 50 to 1000 amu), combined with MS$^E$ centroid mode, including 2 independent scans with different collision energies (CE), alternatively acquired during the run. Parameters of high resolution mass spectrometry included a low-energy scan (CE 4eV) and a high-energy scan (CE ramp 20-35eV) to fragment the ions. Argon (99.999%) was used as collision-induced dissociation gas. ESI conditions comprised of scan rate 0.2 s/scan, capillary voltage 2 kV (negative mode) and 3 kV (positive mode), reference capillary voltage 2.5 kV, cone voltage 40 V, source offset 80 V, source temperature 120 ˚C, desolvation gas temperature 450 ˚C, desolvation gas flow 900 L/h and cone gas flow 50 L/h. Nitrogen (>99.5%) was employed as desolvation and cone gas. The QCs were injected at regular intervals throughout the analytical run to provide a set of data from which repeatability could be assessed.

## Data preprocessing

UNIFI 1.8.1 Software was used for the collection of raw data (Waters Corporation, Milford, USA). The acquired LC-MS raw data were analyzed by the progenesis QI software (Waters Corporation, Milford, USA). Precursor tolerance was set as 5 ppm, product tolerance was set as 10 ppm and retention time (RT) tolerance was set at 0.02 min. Internal standard detection parameters were deselected for peak RT alignment and isotopic peaks were excluded for analysis. Noise elimination level was set at 10.00; minimum intensity was set to 15% of base peak intensity. The Excel file was obtained with three dimension data sets including m/z. Peak RT, peak intensities and RT–m/z pairs were used as the identifier for each ion. The resulting matrix was further reduced by removing any peaks with missing value (ion intensity = 0), in more than 60% samples. The internal standard was used for data QC (reproducibility). Compounds were identified on the basis of accurate mass, secondary fragmentation and isotopic distribution, using the Human Metabolome Database (HMDB), Lipidmaps 2.3 and the METLIN database. The positive and negative ion data were combined into one data matrix table containing all the information that could be used for analysis. The Pathway Analysis was performed on significantly altered known metabolites by using *Arabidopsis thaliana* as the pathway library to

associate the biological functions of identified metabolites to different pathways. Metabolomics data was subjected to Kyoto Encyclopedia of Genes and Genomes (KEGG; http://www.genome.jp/kegg).

## Statistical analysis

For the identification of metabolites showing differential response, they were subjected to statistical analysis using Progenesis IQ software. The data matrix was imported into the SIMCA software package 14.0 (Umetrics, Umeå, Sweden), using unsupervised principal component analysis (PCA), to observe the overall distribution between samples and stability of the entire analysis process. To distinguish the overall differences in metabolic profiles between groups and to find differential metabolites between groups, orthogonal partial least squares discriminant analysis (OPLS-DA) and partial least squares discriminant analysis (PLS-DA) were used. In our study, default 7-round cross-validation was applied with $1/7^{th}$ of the samples being excluded from the mathematical model in each round, in order to guard against over fitting. Data was subjected to two-tailed Student's t test and fold change analysis. The volcano plot was used to visualize p-value and fold change value, which is useful for screening differential metabolites.

A combination of multidimensional analysis and single-dimensional analysis was used to screen differential metabolites between groups. The differential metabolites were selected on the basis of the combination of a statistically significant threshold of variable influence on projection (VIP) values, obtained from the OPLS-DA model and p values from a two-tailed Student's t test on the normalized peak areas. Metabolites with VIP values larger than 1.0 and p values less than 0.05 were considered as differential metabolites. Wherein, the change factor (fold change) was the ratio of the average content of metabolites in two groups and the mass error cut-off was set at <5ppm.

For the identification of differential metabolites among four datasets, Venn diagram was constructed using R software. A fold change threshold of FC >1 was applied to the significant metabolites (DEGS). Heat maps were constructed using MetaboAnalyst 3.0 software. (www.metaboanalyst.ca).

## Results

### Disease severity

Inoculated cotton leaves were examined for disease severity after 4 days of inoculation. The resistant variety exhibited smaller leaf spots of about 2 mm average size and these spots didn't spread further (Figs 1A and 2). Susceptible variety revealed leaf spots of 1.4 cm average size, rapidly, followed by necrosis. Leaf spots spread on the entire leaves, after 3 to 4 days of inoculation. Initially, leaf spot symptoms appeared along the veins of leaves, forming small brownish irregular spots, which gradually increases in size and number and eventually lead to necrosis. Brown spots were more prevalent in the middle of the leaf (Fig 1B). These symptoms were similar to already reported Aspergillus leaf spot [13, 26]. No disease was observed in mock inoculated leaves of both varieties (Fig 1C and 1D).

### Metabolite profiling in response to *A. tubingensis*

Data was obtained in the form of LCMS chromatograms, which are the functions of their retention time and mass to charge ratio. Out of 20202 original peaks for positive ion sample, 17196 peaks were reserved, indicating a yield of 85.13%. Similarly, out of 17713 original peaks for negative ion sample, 15477 peaks were reserved, indicating a yield of 87.38%. As a result of

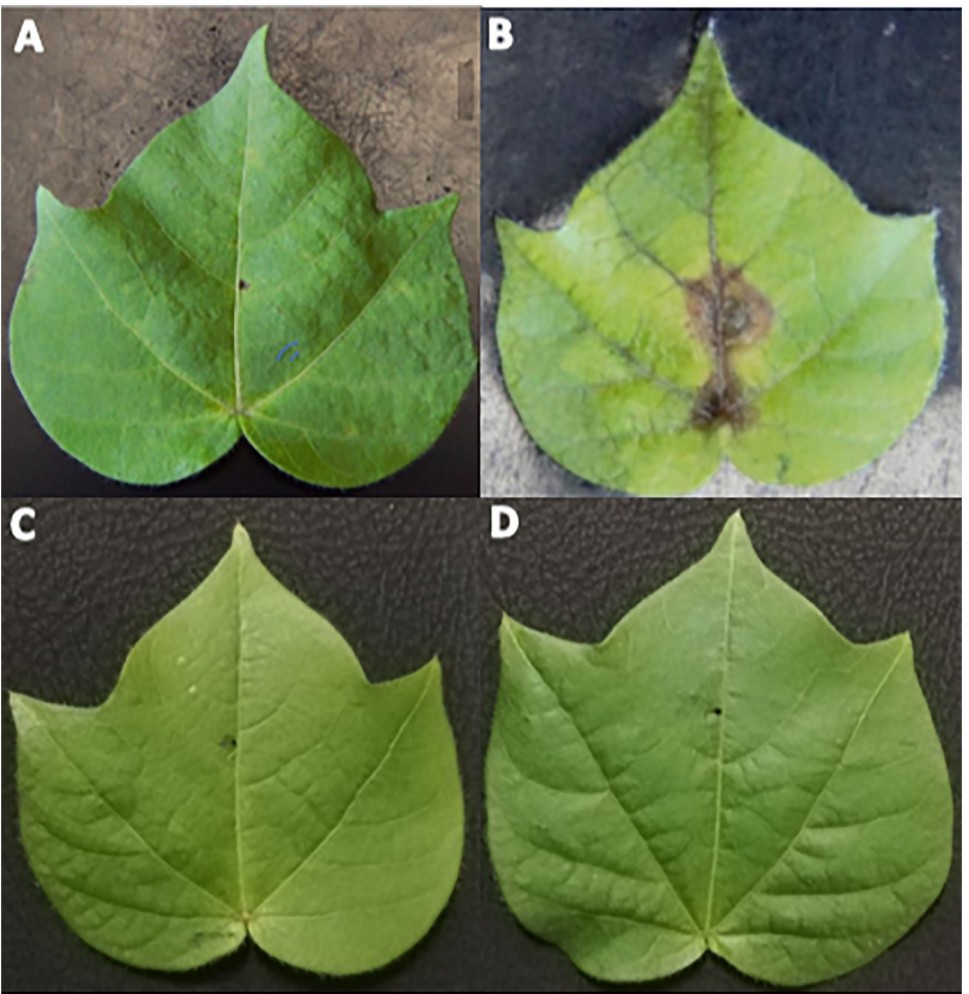

**Fig 1. Cotton leaf spot caused by *A. tubingensis.*** (A) NIA- Sadori (resistant variety) (B) CIM-573 (susceptible variety) (C) NIA-Sadori (mock inoculated) (D) CIM-573 (mock inoculated).

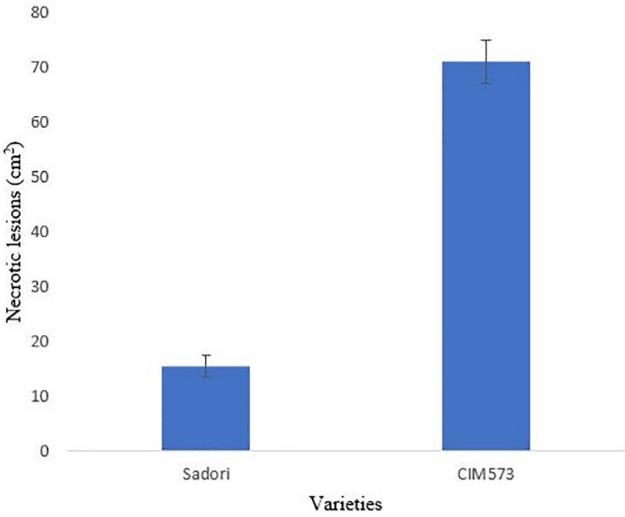

**Fig 2. Measurement of disease severity in $V_1$ = CIM-573 variety and $V_2$ = NIA-Sadori variety.**

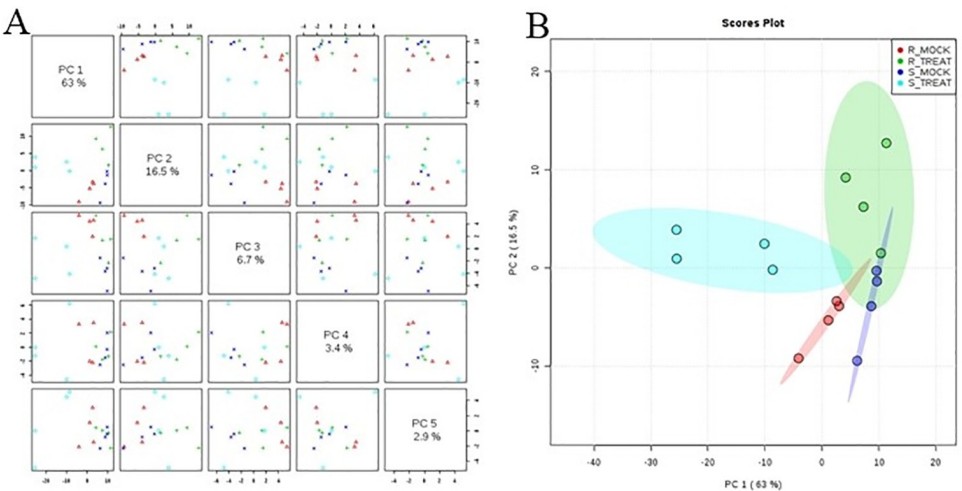

**Fig 3. Principal Component analysis (A) and 2D Scores plot (B) in cotton leaves under control (mock) and treated (pathogen inoculated) conditions.** Mock and treated samples formed separate groups, indicating an altered state of metabolite levels in the leaves. Slight overlapping with each other was also observed.

metabolomics by untargeted LCMS, 32674 peaks were obtained; out of which 7821 metabolites were putatively identified.

## Statistical analysis

Unprocessed data was subjected to multivariate and univariate analysis to confirm the stability and repetition of our experimental work. Multivariate statistical analysis revealed noticeable differences between the samples under mock treatment (RM and SM) and pathogen inoculated treatment (RT and ST). Principle Component Analysis (PCA) demonstrated variance between the samples. Each variety formed its own cluster of metabolites with slight overlapping with each other. PC1 and PC2 showed variance of 63% and 16%, respectively (Fig 3A and 3B). OPLS data analysis indicated the variation between samples of mock and inoculated treatment. The OPLS-DA score plots illustrated the evidences of variation between two varieties of cotton under mock and pathogen inoculated treatments (Fig 4). The PLSDA score plots of our

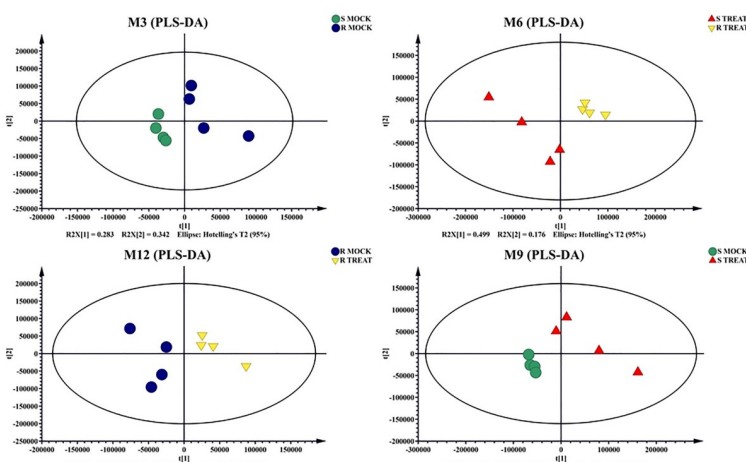

**Fig 4. OPLS-DA plots of four data sets: Resistant Mock vs Susceptible Mock, Resistant Treated vs Susceptible Treated, Susceptible Mock vs Susceptible Treated, Resistant Treated vs Resistant Mock.**

study verified the results of RM vs. SM and RM vs. RT, obtained through PCA. Negative relationship of RT and ST metabolites indicated disease tolerance and susceptibility of resistant and susceptible cultivars, respectively (Fig 5).

For univariate analysis, data was subjected to fold change analysis. Positive and negative values indicated up regulated and down regulated metabolites, respectively. For the identification of significant metabolites, Student's t-test was applied to the data. Volcano plots illustrated highly significant metabolites having lower p-values within four datasets i.e. RM vs. SM, RT vs. ST, RT vs. RM and ST vs. SM. Volcano map revealed up and down regulation of metabolites (Fig 6).

## Differential metabolite screening

Statistical analysis revealed the differential response of about 873 metabolites. Out of these, 528 metabolites were found as primary metabolites, secondary metabolites, membrane lipids and various other small organic compounds. Venn diagram demonstrated differential metabolites between different data sets. Overlapping section of Venn diagram exhibited 12 metabolites, common in all four datasets (Fig 7).

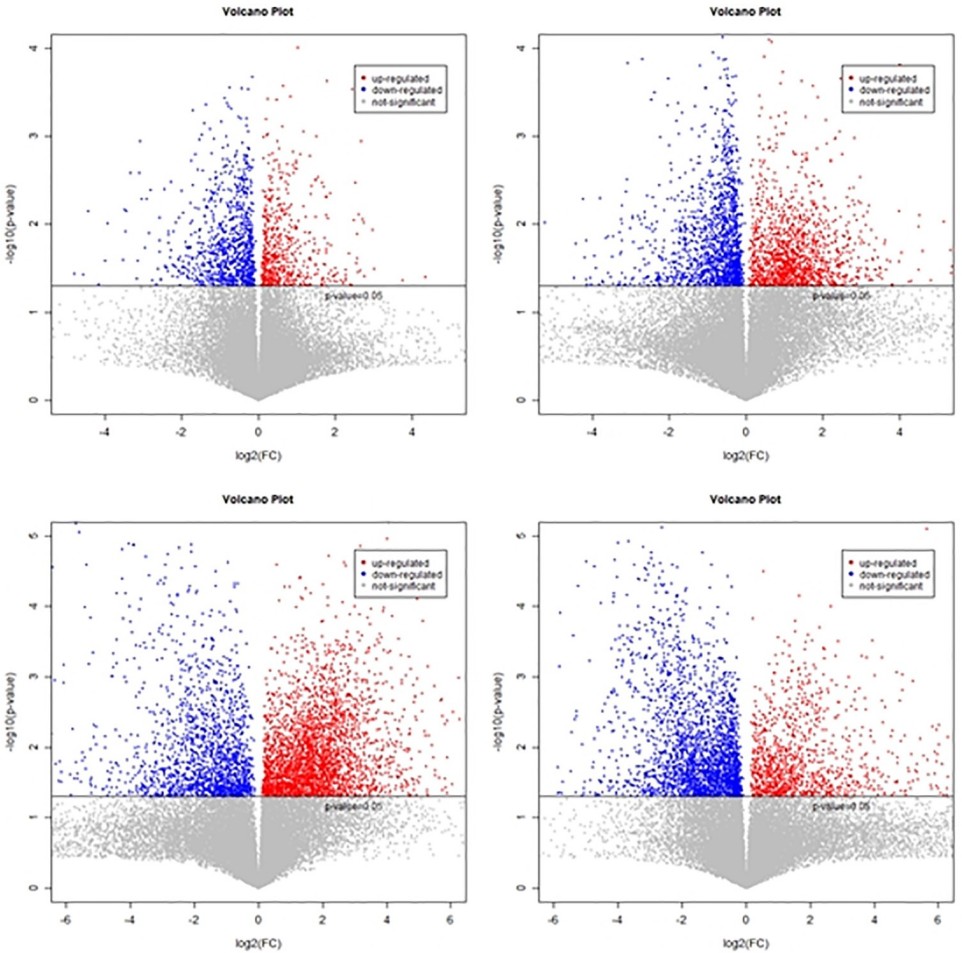

**Fig 5. PLS-DA of four data sets: Resistant Mock vs Susceptible Mock, Resistant Treated vs Susceptible Treated, Susceptible Mock vs Susceptible Treated, Resistant Treated vs Resistant Mock.**

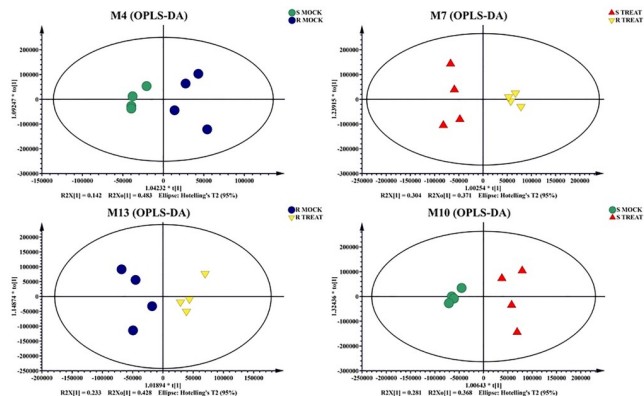

**Fig 6. Volcano maps of four data sets: Resistant Mock vs Susceptible Mock, Resistant Treated vs Susceptible Treated, Susceptible Mock vs Susceptible Treated, Resistant Treated vs Resistant Mock.**

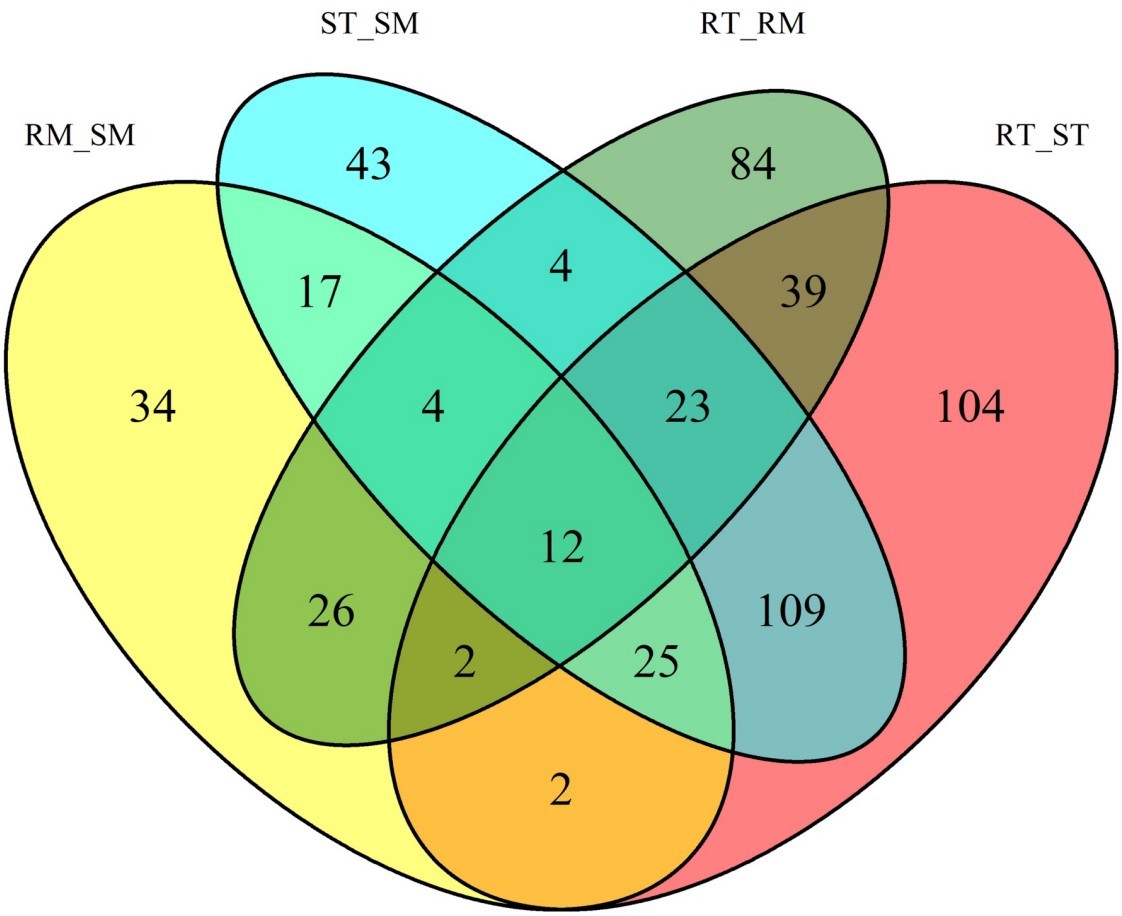

**Fig 7. Venn diagram of differential metabolites in cotton leaf, after pathogen infection.**

## Metabolite profiling of cotton varying in leaf spot resistance

Resistance related (RR) metabolites were identified in the resistant variety to explore the basis of resistance. RR metabolites were further divided into two categories i.e. Resistance related constitutive (RRC) and Resistance related induced (RRI) metabolites. By applying FC >1 cut-off, 241 metabolites were found to be RR metabolites. Out of these, 18 metabolites were identified as RRC (S1 Table) and 223 were observed as RRI metabolites (S2 Table). Most of these RR metabolites exhibited higher values of fold change and were found abundantly in the resistant cultivar NIA-Sadori, as compared to the susceptible cultivar CIM-573. Identification and classification of metabolites was performed from Human Metabolome Database (HMDB), Lipids Map and METLIN Database. Metabolites which were specifically found interfering with plant physiology mainly belonged to carbohydrates, amino acids, flavonoids, phenylpropanoid, alkaloids, terpenoids, steroids, fatty acids and organic acids (Table 1). Heat maps of RRC and RRI metabolites were drawn using MetaboAnalyst (Figs 8 and 9).

## Changes in plant metabolic pathways using KEGG pathway analysis

The Pathway Analysis was performed on significantly altered known metabolites by using *A. thaliana* as the pathway library to associate the biological functions of identified metabolites to different pathways. Three pathways which were common in both varieties (NIA-Sadori and CIM-573) were Alanine, aspartate, glutamate metabolism, glutathione metabolism and Aminoayl tRNA biosynthesis. Both primary and secondary metabolic pathways were disturbed in cotton leaves upon infection with pathogen. Pathways which were significantly altered in resistant variety upon inoculation were citrate cycle, pyruvate metabolism, flavonoid, glyoxylate and dicarboxylate, Biosynthesis of alkaloids, flavone and flavonol biosynthesis, arginine and proline metabolism, histidine metabolism, nitrogen metabolism and energy metabolism. While in susceptible cultivar, arachidonic acid, cyanoamino acid and glycerophospholipid metabolism pathways were changed (Table 2).

## Discussion

In this experiment, based on UPLC Q-Tof technology platform, combined with QI metabolomics data processing software, metabolic profile analysis was performed on cotton varieties. The quality control results showed that the QC samples were gathered together, which indicates the stability and reliability of the whole operation and the experimental platform. Differences in metabolic profiles obtained in the experiments reflect biological differences between samples. Multivariate statistical analysis and t-test were used to screen differential metabolites. Our results showed that there were significant differences in metabolites in different treatment groups. By analyzing diverse metabolome of two cotton varieties (resistant NIA-Sadori and susceptible CIM- 573), under mock treated and pathogen infected conditions, significant accumulation of key metabolites were observed. Identified significant important metabolites mainly belonged to carbohydrates, fatty acids, steroids, terpenoids, flavonoids, alkaloids, phenylpropanoids, amino acids and organic acids.

Higher accumulation of sugars under pathogen infection was observed in this study, which clearly shows the potential for an advanced tolerance during biotic stress in cotton plant. The extended level of carbohydrates like Inulobiose in resistant variety is evident in evading stress. Inulobiose has been reported for its antioxidant activity in chicory [27]. Organic acids like L-Malic acid and Pyruvic acid were highly accumulated in resistant cultivar, in response to pathogen inoculation. Up-regulation of primary metabolism plays a role in signal transduction during stress and modulates the defense response. Pyruvic acid is involved in the induction of hypersensitive response (HR) in plants, leading to the accumulation of reactive oxygen species (ROS) and

**Table 1. List of few significantly important metabolites (RRI and RRC) with their names, molecular formula (MF), compound type, fold change (FC) and genomes identifier number (KEGG ID/HMDB ID) in cotton leaves under control and pathogen inoculated conditions.**

| Metabolite Name | KEGG/HMDB ID | Molecular Formula | Compound Type | Fold changes in SMRM RTST STSM RTRM | | | |
|---|---|---|---|---|---|---|---|
| | | | | SMRM | RTST | STSM | RTRM |
| Glutathione | C00051 | $C_{10}H_{17}N_3O_6S$ | Amino acid | | | 2.7 | 3.46 |
| Pyruvic acid | C00022 | $C_3H_4O_3$ | Organic acid | | 4.68 | | |
| L-Malic acid | C00149 | $C_4H_6O_5$ | Organic acid | | 3.81 | | |
| Prostaglandin | C00584 | $C_{23}H_{38}O_8$ | Fatty acyls | 0.432 | 3.94 | 0.254 | 2.211 |
| Quercitin | C00389 | $C_{15}H_{10}O_7$ | Flavonoids | | 1.50 | | |
| L-Asparagine | C00152 | $C_4H_8N_2O_3$ | Amino acid | | 0.17 | 60.96 | |
| Phosphocholine (PC) | C00588 | $C_5H_{14}NO_4P$ | Phospholipid | 0.566 | | 3.45 | 2.151 |
| Petunidin | C08727 | $C_{16}H_{13}ClO_7$ | Flavonoid | | | 0.50 | |
| (-)-Epigallocatechin | C12136 | $C_{15}H_{14}O_7$ | Flavonoid | | | 0.48 | |
| (+)-Gallocatechin | C12127 | $C_{15}H_{14}O_7$ | Flavonoid | | 2.04 | 0.477 | |
| Glycyrol | C16968 | $C_{21}H_{18}O_6$ | Isoflavonoid | | 3.5 | 0.3 | |
| Aloinoside B | C17780 | $C_{27}H_{32}O_{13}$ | Benzenoid | | 3.05 | 0.36 | |
| Phlorizin | C01604 | $C_{21}H_{24}O_{10}$ | Flavonoid | | 3.39 | 0.24 | |
| Aspidofractine | C09040 | $C_{22}H_{26}N_2O_3$ | Alkaloid | | 5.83 | 0.31 | |
| Asparagoside A | C08886 | $C_{33}H_{54}O_8$ | Steroid | | 3.2 | 0.31 | |
| Dioscin | C08897 | $C_{45}H_{72}O_{16}$ | Steroid | | 10.5 | 0.12 | 1.96 |
| Bis(glutathionyl)spermine | C16563 | $C_{30}H_{56}N_{10}O_{10}S_2$ | Amino acid | | | 0.25 | |
| Spinasaponin A | C08984 | $C_{42}H_{66}O_{14}$ | Terpenoid | 0.44 | 4.7 | 0.17 | 1.88 |
| L-Fucose | C01019 | $C_6H_{12}O_5$ | Sugar | | 0.466 | | |
| Quercetin 3-O-glucoside | C05623 | $C_{21}H_{20}O_{12}$ | Flavonoid | | 1.64 | | |
| Orientin | C10114 | $C_{21}H_{20}O_{11}$ | Flavonoid | | 1.88 | | |
| Cellobiose | C00185 | $C_{12}H_{22}O_{11}$ | Sugar | | 3.4 | | |
| Chrosimate | C00251 | $C_{10}H_{10}O_6$ | Organic acid | | 0.4 | | |
| Baptifoline | C10755 | $C_{15}H_{20}N_2O_2$ | Alkaloid | | 0.39 | | |
| Citramalic acid | C00815 | $C_5H_8O_5$ | Fatty acid | 1.73 | | | |
| Tragopogonsaponin C | HMDB0037912 | $C_{51}H_{72}O_{17}$ | Terpenoid | | 6.98 | | |
| Azl | HMDB0031775 | $C_{48}H_{74}O_{18}$ | Terpenoid | | 6.81 | 0.21 | 2.28 |
| Flazine | HMDB0033459 | $C_{17}H_{12}N_2O_4$ | Alkaloid | | 2.10 | 0.47 | |
| Benzosimuline | HMDB0031930 | $C_{20}H_{19}NO_2$ | Alkaloid | | 2.32 | | |
| Methionyl-Phenylalanine | HMDB0028980 | $C_{14}H_{20}N_2O_3S$ | Amino acid | | 2.30 | 0.32 | |
| Acetylcorinoline | 73548 | $C_{23}H_{23}NO_6$ | Alkaloid | | 2.30 | 0.37 | |
| Torvoside F | HMDB0041531 | $C_{45}H_{74}O_{18}$ | Steroid | | 8.95 | 0.2 | 2.33 |
| Alliofuroside A | HMDB0041051 | $C_{44}H_{72}O_{18}$ | Steroid | | 10.98 | | |
| Olitorin | HMDB0034361 | $C_{35}H_{52}O_{14}$ | Steroid | | | | 2.4 |
| Cyclopassifloside VII | HMDB003 | $C_{37}H_{62}O_{13}$ | Terpenoid | | 2.44 | 0.36 | |
| Oroselone | HMDB0033925 | $C_{14}H_{10}O_3$ | Phenylpropanoid | | 2.4 | | |
| Theaflavin | HMDB0005788 | $C_{29}H_{24}O_{12}$ | Flavonoid | | 1.8 | 0.53 | |
| Wampetin | HMDB0030080 | $C_{21}H_{18}O_6$ | Phenylpropanoid | | 3.6 | 0.2 | |
| Tuberoside D | HMDB0034304 | $C_{45}H_{74}O_{17}$ | Steroid | | | | 3.6 |
| Torvoside C | HMDB0029623 | $C_{39}H_{64}O_{13}$ | Steroid | 0.33 | 3.62 | 0.24 | 2.62 |
| Camelliagenin B | 90000 | $C_{30}H_{48}O_5$ | Terpenoid | | 5.2 | | |
| Launobine | HMDB0030217 | $C_{18}H_{17}NO_4$ | Alkaloid | | 5.0 | | |
| Goyaglycoside c | HMDB0038349 | $C_{38}H_{62}O_9$ | Steroid | | 4.7 | | |
| Floribundoside | HMDB0033739 | $C_{21}H_{22}O_{10}$ | Flavonoid | | 4.5 | | |
| Pisumoside B | HMDB0037125 | $C_{32}H_{52}O_{16}$ | Terpenoid | | 4.4 | | |
| Bayogenin | 53775 | $C_{30}H_{48}O_5$ | Terpenoid | | 4.2 | | |

*(Continued)*

**Table 1.** (Continued)

| Metabolite Name | KEGG/HMDB ID | Molecular Formula | Compound Type | Fold changes in SMRM | RTST | STSM | RTRM |
|---|---|---|---|---|---|---|---|
| Patuletin | C10118 | $C_{16}H_{12}O_8$ | Flavonoid | 4.1 | | | |
| Divanillyltetrahydrofuran ferulate | HMDB0032730 | $C_{30}H_{32}O_8$ | Phenylpropanoid | 4.0 | | | |
| Dihydropanaxacol | HMDB0032675 | $C_{17}H_{28}O_3$ | Flavonoid | 3.3 | | | |
| N-a-Acetyl-L-arginine | HMDB0004620 | $C_8H_{16}N_4O_3$ | Amino acid | 2.1 | | | |
| Pitheduloside I | HMDB0034036 | $C_{30}H_{48}O_5$ | Terpenoid | 2.9 | | | |
| Asparaginyl-Isoleucine | HMDB0028734 | $C_{10}H_{19}N_3O_4$ | Amino acid | 2.2 | | | |
| Argenteane | HMDB0039454 | $C_{40}H_{46}O_8$ | Lignin | 2.6 | | | |
| 4-Hydroxynonenal | HMDB0036332 | $C_9H_{16}O_2$ | Fatty acid | 3.1 | | | |
| Inulobiose | C01711 | $C_{12}H_{22}O_{11}$ | Sugar | 3.18 | | | |
| Isolicopyranocoumarin | HMDB0035479 | $C_{21}H_{20}O_7$ | Flavonoid | 3.1 | | | |
| Hoduloside III | HMDB0039072 | $C_{47}H_{76}O_{17}$ | Terpenoid | 3.3 | | | |
| Molludistin 2"-rhamnoside | HMDB0037418 | $C_{27}H_{30}O_{13}$ | Flavonoid | 3.4 | | | |
| alpha-Spinasterol 3-glucoside | HMDB0033775 | $C_{35}H_{58}O_6$ | Steroid | 3.6 | | | |
| 4'-Hydroxyacetophenone 4'-[4hydroxy-3,5-dimethoxybenzoyl(->5)-apiosyl-(1->2)-glucoside] | HMDB0036332 | $C_{28}H_{34}O_{15}$ | Tannin | 3.7 | | | |
| Myricetin 3-[glucosyl-(1->2)rhamnoside] 7-[rhamnosyl-(1>2)-glucoside] | HMDB0038823 | $C_{39}H_{50}O_{26}$ | Flavonoid | 4.00 | | | |
| Nelumboside (RRC) | HMDB0038464 | $C_{27}H_{28}O_{18}$ | Flavonoid | 3.2 | | | |
| Cyclotricuspidoside C (RRC) | HMDB0033636 | $C_{43}H_{72}O_{17}$ | Terpenoid | 2.4 | | | |
| Aromadendrin (RRC) | C00974 | $C_{15}H_{12}O_6$ | Flavonoid | 1.8 | | | 0.6 |
| Stigmasterol (RRC) | C05442 | $C_{29}H_{48}O$ | Fatty acid | 1.4 | | | 0.67 |
| Delta 2- THA (δ2-tetracosahexaenoic acid) (RRC) | LMFA01030852 | $C_{24}H_{34}O_2$ | Fatty acid | 1.12 | | | |
| (3b,9R)-5-Megastigmene-3,9-diol 9-[apiosyl-(1->6)glucoside] (RRC) | HMDB0038327 | $C_{24}H_{42}O_{11}$ | Fatty acid | 1.95 | | | |
| Elaterinide (RRC) | HMDB0035893 | $C_{38}H_{54}O_{13}$ | Steroid | 1.9 | | | |
| 2,2,4,4,-Tetramethyl-6-(1-oxopropyl)-1,3,5cyclohexanetrione | HMDB0033191 | $C_{13}H_{18}O_4$ | Terpenoid | 1.08 | | | |
| L-Glutamate | C00025 | $C_5H_9NO_4$ | Amino acid | | 2.6 | | 2.5 |
| L-Aspartic acid | C00049 | $C_4H_7NO_4$ | Amino acid | | | | 2.5 |
| N2-Fructopyranosylarginine | HMDB0041541 | $C_{12}H_{24}N_4O_7$ | Amino acid | | 26.9 | | |
| Cinnzeylanol | HMDB0036010 | $C_{20}H_{32}O_7$ | Terpenoid | | 15.6 | | 2.8 |
| Calenduloside B | HMDB0039413 | $C_{48}H_{78}O_{18}$ | Terpenoid | | 14.1 | 0.08 | |
| (-)-Dioxybrassinin | HMDB0038634 | $C_{11}H_{12}N_2O_2S_2$ | Amino acid | | 9.6 | | |
| Dihydrowyerone | HMDB0039493 | $C_{14}H_{14}O_4$ | Fatty acid | | 1.2 | 0.6 | 0.6 |
| N-a-acetylcitrulline | HMDB000856 | $C_8H_{15}N_3O_4$ | Amino acid | | 2.1 | | 0.4 |
| Raphanusamic acid | HMDB0041280 | $C_4H_5NO_2S_2$ | Amino acid | | 0.3 | | |
| Epirosmonol | HMDB0035812 | $C_{20}H_{26}O_5$ | Terpenoid | | | | 2.8 |
| Arjunolic acid | HMDB0034502 | $C_{30}H_{48}O_5$ | Terpenoid | | | | 1.9 |
| Oleanolic acid | HMDB0036357 | $C_{41}H_{66}O_{12}$ | Terpenoid | | 2.8 | 7.3 | 0.2 |
| Cephradione A | HMDB0034364 | $C_{18}H_{11}NO_4$ | Alkaloid | | 1.7 | 0.5 | |

Where SMRM (Susceptible mock inoculated, Resistant mock inoculated) RTST (Resistant treated and Susceptible treated) STSM (Susceptible treated and mock inoculated) RTRM (Resistant treated and mock inoculated)

Compound Database ID = HMDB: Human Metabolome Database, LMGP and LMFA: LIPID MAPS, Number: METLIN.

RRI: RT vs ST, RT vs RM, ST vs SM.

RRC: RM vs SM

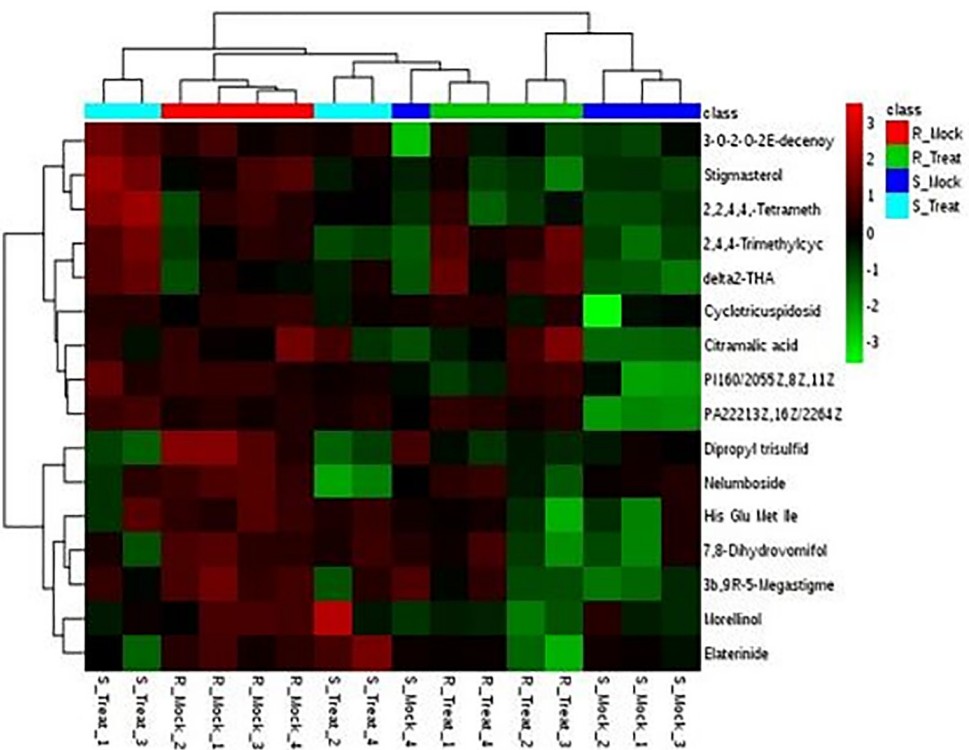

**Fig 8. Hierarchial clustering showing heat map of RRC metabolites with fold change > 1 in the Mock treatments (R_Mock vs. S_Mock) and their response in other treatments generated using Metaboanalyst software.** Red and green colors represent up and down regulation, respectively. Columns are exhibiting samples and rows are exhibiting metabolites here.

in turn, up-regulation of pathogenesis related (PR) proteins [28]. L-Malic acid is an intermediate of TCA (tricarboxylic acid) cycle and its efflux from plant roots acts as a signal for recruiting beneficial rhizobacteria [29]. The accumulation of some organic acids including citric acid could contribute to greater capacity of some genotype of cotton to manage drought stress [30].

Amino acids with relative higher accumulation in resistant cultivar included Dioxibrassinin, N2-Fructopyranosylarginine, Glutathione and L-Glutamate. Dioxibrassinin is a phytoalexin that has been reported for its antimicrobial activity against *Bipolaris leersiae* [31]. N2-Fructopyranosylarginine has been described to possess antioxidant properties [32]. Glutathione shows antioxidant activity in plants and regulates the responses of plants to various biotic and abiotic stresses by producing phytoalexins [33]. L-Glutamate plays a key role in amino acid metabolism and signaling during stress [34]. In this study, N-arachidonoyl alanine, L-Aspartic acid, Dihydrowyerone acid and 4-Hydroxynonenal were abundantly accumulated fatty acids in resistant cultivar. N-arachidonoyl alanine has been identified as RR metabolite in wheat for inducing Fusarium head blight (FHB) resistance by acting as physical barriers as well as antimicrobial agents [35]. L-Aspartic acid or aspartate leads to the production of amino acids like asparagine which is employed for nitrogen storage in plants [36]. Dihydrowyerone acid is a phytoalexin, [37] and 4-Hydroxynonenal is an end product of lipid peroxidation [38]. Stigmasterol was also accumulated in mock inoculated resistant variety, which is an intrinsically existing secondary compound in cotton plant [4]. Elevated levels of fatty acids in resistant cultivar under stress conditions are in accordance with previous findings which suggest their role in signal transduction [39].

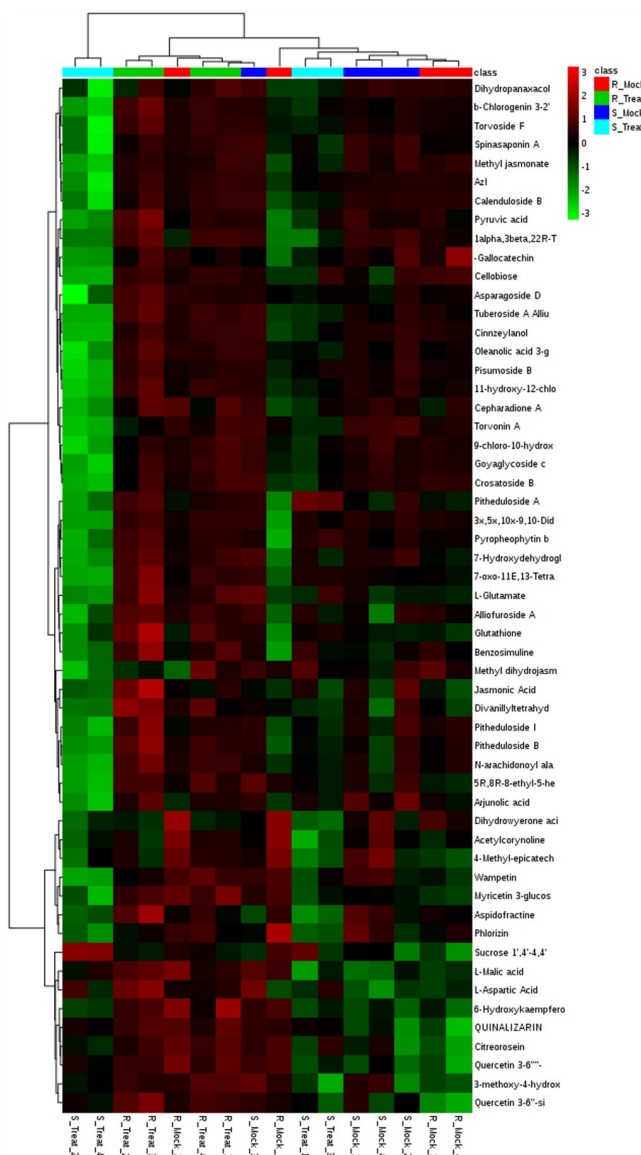

**Fig 9. Hierarchial clustering showing heat map of top 60 RRI metabolites of resistant and susceptible varieties, induced by the infection of *A. tubingensis*, generated using Metaboanalyst software.** Red and green colors represent up and down regulation, respectively. Columns are presenting samples and rows are exhibiting metabolites here. Clustering is evident from the shown dendrograms.

Primary metabolism has also been reported to be up regulated during plant-pathogen interaction. This up regulation of primary metabolic pathways results in cascades of signal transduction in plants under stress. This signaling helps to modulate defense related responses [28]. Plants having higher secondary metabolite content are generally considered to be more resistant against stresses [40]. In this study, various secondary metabolites (phenylpropanoids) including diarylheptanoids, stilbenes, wampetin and oroselone exhibited elevated accumulation in resistant cultivar. Phenylpropanoids are known for their antioxidant properties [41]. In our study, several phytoalexins like Oroselone (furanocoumarin) were induced abundantly in tolerant cotton cultivar and could be considered as potential defense against *A. niger*. Different flavonoids like phlorizin, orientin, quercetin, kaempferol and isoflavanoids were also induced

**Table 2. Pathway ID and names, total metabolites involved in those pathways, metabolites significantly accumulated in present study and false discovery rate (FDR) in cotton leaves of resistant and susceptible variety, identified by pathway analysis using *Arabidopsis thaliana* as the pathway library.**

| ID Annotation | Pathway Annotation | In set | In background | Present in | FDR correction |
|---|---|---|---|---|---|
| ath00940 | Flavonoid biosynthesis | 2 | 68 | SM ST, RT ST, RM SM | 5.1 |
| ath00250 | Alanine, aspartate and glutamate metabolism | 3 | 24 | RT ST, RM RT, SM ST | 1.1 |
| ath00910 | Nitrogen metabolism | 1 | 26 | RM RT, RT ST | 1.1 |
| ath00460 | Cyanoamino acid metabolism | 1 | 41 | SM ST | 1.1 |
| ath00564 | Glycerophospholipid metabolism | 1 | 46 | SM ST | 1.1 |
| ath00970 | Aminoacyl-tRNA biosynthesis | 1 | 75 | RM RT, SM ST | 1.4 |
| ath00590 | Arachidonic acid metabolism | 1 | 75 | SM ST | 1.4 |
| ath00480 | Glutathione metabolism | 4 | 38 | RT ST, SM ST, RM SM, RM RT | 1.3 |
| ath001064 | Biosynthesis of alkaloids derived from ornithine, lysine and nicotinic acid | 3 | 67 | RT ST | 4.0 |
| ath00020 | Citrate cycle (TCA cycle) | 2 | 20 | RT ST | 4.0 |
| ath00710 | Carbon fixation in photosynthetic organisms | 2 | 23 | RT ST, RM RT | 4.3 |
| ath00660 | C5-Branched dibasic acid metabolism | 2 | 32 | RT ST | 4.9 |
| ath00620 | Pyruvate metabolism | 2 | 32 | RT ST | 2.9 |
| ath00944 | Flavone and flavonol biosynthesis | 2 | 33 | RT ST | 4.9 |
| ath00650 | Butanoate metabolism | 2 | 40 | RT ST | 5.9 |
| ath00630 | Glyoxylate and dicarboxylate metabolism | 2 | 44 | RT ST | 6.6 |
| ath0040 | Pentose and glucuronate interconversions | 2 | 53 | RT ST | 7.6 |
| ath00270 | Cysteine and methionine metabolism | 2 | 56 | RM RT, RT ST | 7.6 |
| ath003330 | Arginine and proline metabolism | 2 | 82 | RT ST, RM RT | 1.3 |
| ath00860 | Porphyrin and chlorophyll metabolism | 3 | 126 | RM RT, RT ST | 6.8 |
| ath00340 | Histidine metabolism | 2 | 44 | RM SM, RM RT | 6.9 |
| ath00260 | Glycine, serine and Threonne metabolism | 2 | 49 | RM RT | 7.1 |
| ath01062 | Biosynthesis of terpenoids and steroids | 2 | 98 | RM SM | 1.4 |

abundantly after pathogen inoculation in resistant cultivar. These have been proposed to be involved in disease resistance against various pathogens [42, 43, 44, 45, 46, 47].

Steroids do not have role in plant growth and are primarily involved in defense response of plants against several types of stresses. In resistant variety, high accumulation of steroidal glycosides like Melongoside O, Asparagoside D and A, Alliofuroside A, Sarasapogenin, Schidigerasaponin (F2 and C2), Tuberosides, Torvonin, Alphaspinasterol 3-glucoside and Olitorin suggest their role in plant defense and response to wounding [48]. Resistant cultivar infected with pathogen also depicted higher accumulation of alkaloids including 7-Hydroxydehydroglaucine, Indolizine, Benzosimuline, Acetylcorynoline, Flazine, Benzosimuline and Cepharadione A. These compounds have been reported for their antifungal activities [49, 50, 51, 52]. In this study, numerous membrane glycerolipids including several PE (Phosphatidylethanolamine), PC (Phosphatidylcholine), PI (Phosphatidylinositol), PA (Phosphatidic acid), PIP (Phosphatidylinositol Phosphate), PS (Phosphatidylserine) and phosphocholine were also identified. All these are membrane lipids and function in signaling, in response to various environmental factors such as drought, change in temperature and salinity as well as several biotic stresses [53].

In current study, 23 pathways were demonstrated, out of which Glutathione pathway was commonly altered in both varieties. Glutathione biosynthesis is of significant importance as glutathione plays strong role in scavenging of ROS. It interacts with hormones and signaling

molecules and its redox state triggers signal transduction [54]. Glutathione modulates cell pro-liferation, apoptosis, fibrogenesis, growth, development, cell cycle, gene expression, protein activity and immune function [55]. The biosynthetic pathways of some amino acids such as alanine, aspartate and glutamate, flavonoid biosynthesis and aminoacyl tRNA synthesis were also varied in both varieties. Flavonoids have been found to be accumulated during environmental stresses and protect plant cells through the inhibition of destructive ROS [56].

Citrate cycle (TCA cycle), Glyoxylate, dicarboxylate and Pyruvate metabolism were elevated in resistant variety upon fungal infection. TCA cycle is an essential metabolic pathway which creates energy for different biological activities and also provides precursors used in many biosynthetic pathways [57]. Pyruvate is a key intersection in the network of metabolic pathways [58]. Nitrogen metabolism, cysteine and methionine metabolism, arginine and proline metabolism, porphyrin and chlorophyll metabolism, histidine metabolism and Flavone metabolism were also high in resistant variety on treatment with fungus. Arginine has been reported to accumulate under stress and deficiency conditions and it acts as a precursor of polyamines [59]. Previous findings show that under stress condition, the mitochondrial oxidative phosphorylation is decreased and the yield of ATP is increased through proline metabolic pathway to restore stress induced damage [60]. Flavonols are considered to be the most important flavonoids participating in stress responses; having a wide range of potent physiological activities [61]. Histidine metabolism was also higher in resistant variety under pathogen treated condition. Histidine (His) is one of the standard amino acids in proteins, and plays a critical role in plant growth and development [62].

## Conclusion

Metabolomics analysis of cotton leaves revealed dynamic accumulation of different metabolites, in response to the inoculation of *A*. *tubingensis*. Some of the metabolites were significantly changed only in resistant variety while some were altered in both varieties. Findings of this study helped us to conclude that the accumulation of different kinds of carbohydrates, fatty acid, amino acids, organic acids and flavonoids infer resistance to cotton plant against *A*. *tubingensis* by providing energy and signaling molecules for secondary metabolism. Moreover, the inoculation of *A*. *tubingensis* affects primary metabolism by the up regulation of pyruvate and malate and by the accumulation of carbohydrates like cellobiose and inulobiose. Several RRI metabolites, identified in the present study are the precursors for many secondary metabolic pathways. Suppression of these secondary metabolites in the susceptible variety resulted in the development of disease and their presence in resistant cultivar halted the growth of fungus. Further studies are required to involve transcriptional and genetic analyses to elucidate the pathways involved in defense mechanism of cotton plant.

## Supporting information

**S1 Table. List of resistance related constitutive (RRC) metabolites.**
(XLSX)

**S2 Table. List of resistance related induced (RRI) metabolites.**
(XLSX)

## Acknowledgments

We are thankful to Higher Education Commission (HEC), Pakistan for providing financial support under international research support initiative program (IRSIP-39-BMS-41). We are also grateful to Dr. Muhammad Ahmed for providing cotton seeds.

## Author Contributions

**Conceptualization:** Muhammad Farooq Hussain Munis.

**Formal analysis:** Maria Khizar, Jianxin Shi, Sadia Saleem, Muhammad Ashraf.

**Investigation:** Maria Khizar, Hassan Javed Chaudhary, Muhammad Farooq Hussain Munis.

**Methodology:** Fiza Liaquat.

**Project administration:** Muhammad Farooq Hussain Munis.

**Resources:** Jianxin Shi, Muhammad Farooq Hussain Munis.

**Software:** Maria Khizar, Fiza Liaquat, Syed Waqas Hassan, Shafiq ur Rehman, Umar Masood Quraishi.

**Supervision:** Jianxin Shi, Muhammad Farooq Hussain Munis.

**Validation:** Jianxin Shi, Muhammad Farooq Hussain Munis.

**Visualization:** Maria Khizar, Sadia Latif.

**Writing – original draft:** Maria Khizar, Urooj Haroon.

**Writing – review & editing:** Maria Khizar, Fiza Liaquat, Sadia Latif, Urooj Haroon.

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
