## [Decision Letter · Decision Letter 0]

25 Nov 2019

PONE-D-19-26966

Resistance Associated Metabolite Profiling of Aspergillus Leaf Spot in Cotton through Non-targeted Metabolomics

PLOS ONE

Dear Dr. Munis,

Thank you for submitting your manuscript to PLOS ONE. After careful consideration, we feel that it has merit but does not fully meet PLOS ONE’s publication criteria as it currently stands. Therefore, we invite you to submit a revised version of the manuscript that addresses the points raised during the review process.

We would appreciate receiving your revised manuscript by Jan 09 2020 11:59PM. To enhance the reproducibility of your results, we recommend that if applicable you deposit your laboratory protocols in protocols.io, where a protocol can be assigned its own identifier (DOI) such that it can be cited independently in the future. For instructions see: http://journals.plos.org/plosone/s/submission-guidelines#loc-laboratory-protocols

We look forward to receiving your revised manuscript.

Kind regards,

Kandasamy Ulaganathan

Academic Editor

PLOS ONE

Journal Requirements:

1. PLOS requires an ORCID iD for the corresponding author in Editorial Manager on papers submitted after December 6th, 2016. Please ensure that you have an ORCID iD and that it is validated in Editorial Manager. To do this, go to ‘Update my Information’ (in the upper left-hand corner of the main menu), and click on the Fetch/Validate link next to the ORCID field. This will take you to the ORCID site and allow you to create a new iD or authenticate a pre-existing iD in Editorial Manager. Please see the following video for instructions on linking an ORCID iD to your Editorial Manager account: https://www.youtube.com/watch?v=_xcclfuvtxQ

Additional Editor Comments (if provided):

Thank you for submitting your manuscript. It was reviewed by two reviewers, one focusing on your metabolomics work while the other focusing on the cotton genotypes you have selected for the work. There are serious doubts raised on the basis of selection of genotypes. Please justify the selection of genotypes with suitable evidence and answer all questions. These are essential before we could take a final decision on the suitability of your paper for publication.

Reviewers' comments:

Reviewer's Responses to Questions

**Comments to the Author**

1. Is the manuscript technically sound, and do the data support the conclusions?

Reviewer #1: Yes

Reviewer #2: Partly

2. Has the statistical analysis been performed appropriately and rigorously? 

Reviewer #1: Yes

Reviewer #2: No

3. Have the authors made all data underlying the findings in their manuscript fully available?

Reviewer #1: Yes

Reviewer #2: Yes

4. Is the manuscript presented in an intelligible fashion and written in standard English?

Reviewer #1: Yes

Reviewer #2: No

5. Review Comments to the Author

Reviewer #1: The manuscript “Resistance Associated Metabolite Profiling of Aspergillus Leaf Spot in Cotton through Non-targeted Metabolomics” described that the numerous metabolites, identified through ultra high performance liquid chromatography-mass spectrometry (UPLC-MS), are changed in response to the A. tubingensis infection in cotton. It will contribute to a better understanding of the resistant mechanism in both cultivars.

The data is solid and suitable for publication in “PloS One” after a minor revision. Meanwhile, the whole manuscript should be further polished. Introduction should illustrate in-depth background of the manuscript. In discussion section, focus should be on the discussion of the results rather than reference’s accumulation. There are some grammatical and formatting mistakes. Below some minor suggestions are listed:

Line 18. The “and abiotic” should be deleted.

Line 19. The “behavior” is more relevant term to animals, rather than plants.

Line 36. The level is same or different?

Line 39. Full stop should be added at the end of the sentence.

Line 45-53. The authors should focus more on Leaf spot of cotton (Gossypium hirsutum L.) and Aspergillus tubingensis rather than other pathogens of cotton.

Line 69-70. Is this sentence in close relationship with the context?

Line 98-100. The description is unclear, should be explained in a bit detail.

Line 194-200 and Fig. 1 and 2. There are four treatment groups, including RT, RM, SM and ST. The authors have mentioned just two groups in Fig. 1 and 2. They should also include the other two groups. Meanwhile, the leaf spot area in susceptible and resistant varieties is much larger than 5 fold in Fig. 1, while in Fig. 2 it is shown about 5 fold. The ratio in both figures does not match.

Line 337. Reference [23] should be replaced with “Levi et al.” or the sentence should be rephrased and the reference should be placed at the end of the sentence.

Line 382. “are said to” is very colloquial, so rephrase the sentence.

Reviewer #2: The research seems to be original but of not of very practical use. There is no literature available on the incidence and the losses caused due to A. tubigensis anywhere on cotton crop.: There is no information about the pathogen and identification/registration. There is no detail available wrt purity of seed and the authorized source. Methodology followed for the experimentation and inoculation is not clear; There is no reference for resistant (Sadori) and susceptible variety (CIM 88 573). How and on what basis these were selected before conducting the experiment on Gossypium hirsutum. There are no clear cut details and or references about disease symptoms, ratting and AUDPC estimation.

6. PLOS authors have the option to publish the peer review history of their article (what does this mean?). If published, this will include your full peer review and any attached files.

Reviewer #1: No

Reviewer #2: No

---

## [Author Response · Author response to Decision Letter 0]

7 Jan 2020

ANSWERS TO REVIEWERS COMMENTS

Resistance Associated Metabolite Profiling of Aspergillus Leaf Spot in Cotton through Non-targeted Metabolomics

To be accepted for publication in PLOS ONE, research articles must satisfy the following criteria: 

1. The study presents the results of original research. The research seems to be original but of not of very practical use. There is no literature available on the incidence and the losses caused due to A. tubigensis anywhere on cotton crop.

Answer:

In last two seasons, leaf spot disease has become one of the devastating cotton diseases in Pakistan and it was reported in national print newspapers in URDU language. No one reported this disease yet and after intensive field surveys and repeated pathogenicity tests, our lab has already reported these losses in an international journal. Our manuscript has been accepted and will be available online, very soon.

2. Results reported have not been published elsewhere. Need to be checked online. 

Answer:

We do hereby declare that the results of this study have not been submitted or published, anywhere else.

3. Experiments, statistics, and other analyses are performed to a high technical standard and are described in sufficient detail. : There is no information about the pathogen and identification/registration. There is no detail available wrt purity of seed and the authorized source. Methodology followed for the experimentation and inoculation is not clear; There is no reference for resistant (Sadori) and susceptible variety (CIM 573). How and on what basis these were selected before conducting the experiment on Gossypium hirsutum. There are no clear cut details and or references about disease symptoms, ratting and AUDPC estimation. 

Answer:

As mentioned in previous query, pathogen identification and pathogenicity results will be published online, soon. All the seeds were collected from authorized research institute of cotton and we have mentioned it in the text. Methodology followed for the experimentation and inoculation has been described in a detailed manner. References for resistant (Sadori) and susceptible variety (CIM 573) have been incorporated. Clear cut details about disease symptoms, ratting and AUDPC estimation have been added. 

4. Conclusions are presented in an appropriate fashion and are supported by the data. The nontargeted metabolomics approach aims to study both known and unknown metabolites. To comprehend the huge chunks of data this may yield, one must couple it with advanced chemometric methods such as multivariate analysis, so that these can be grouped into smaller manageable chunks. In this article no clear cut and pin pointed conclusions are drawn and the chemicals detected in resistant variety without and with inoculation are mostly of known nature. The discussion and conclusion are very vague and not very relevant to the findings.

Answer: In the revised manuscript, both multivariate analysis and PLSDA have been performed and described. As per instruction, conclusion has been modified as a precise and informative finding for the reader. This study is of significant importance because the reaction of Cotton against Aspergillus tubingensis has never been elaborated before. Both discussion and conclusion have been modified, as per instruction. 

5. The article is presented in an intelligible fashion and is written in standard English. Need English editing, and reference write-up under discussion part need to be shortened. Too much detail is given in the discussion part. Relevant references particularly on G. hirsutum are missing. Reference arrangement and style is not corrects. 

Answer: English editing has been done carefully. Discussion has been revised and shortened to half. References relevant to G. hirsutum have been incorporated. Reference arrangement and style has been corrected. 

ANSWERS TO EDITOR’S COMMENTS

Resistance Associated Metabolite Profiling of Aspergillus Leaf Spot in Cotton through Non-targeted Metabolomics

Reviewer Comments to the author:

Reviewer #1:

 Query 1:

Line 18. The “and abiotic” should be deleted

Answer: Desired changes have been incorporated.

Query 2:

Line 19. The “behavior” is more relevant term to animals, rather than plants.

Answer: Plant relevant term “response” has been added.

Query 3:

Line 36. The level is same or different?

Answer: For better understanding, sentence has been reconstructed.

Query 4:

Line 39. Full stop should be added at the end of the sentence.

Answer: Required changes have been incorporated.

Query 5: 

Line 45-53. The authors should focus more on Leaf spot of cotton (Gossypium hirsutum L.) and Aspergillus tubingensis rather than other pathogens of cotton

Answer: Literature related to Gossypium hirsutum and Aspergillus tubingensis has been added.

Query 6:

Line 69-70. Is this sentence in close relationship with the context?

Answer: As per instruction, desired changes have been made. 

Query 7:

Line 98-100. The description is unclear, should be explained in a bit detail.

Answer: Fungal Inoculation methodology has been explained in detail.

Query 8:

Line 194-200 and Fig. 1 and 2. There are four treatment groups, including RT, RM, SM and ST. The authors have mentioned just two groups in Fig. 1 and 2. They should also include the other two groups. Meanwhile, the leaf spot area in susceptible and resistant varieties is much larger than 5 fold in Fig. 1, while in Fig. 2 it is shown about 5 fold. The ratio in both figures does not match.

Answer: Keeping in view the reviewer’s concerns, pictures of RM and SM have been added. Required changes have been incorporated.

Query 9:

Line 337. Reference [23] should be replaced with “Levi et al.” or the sentence should be rephrased and the reference should be placed at the end of the sentence.

Answer: Desired changes have been incorporated.

Query 10:

Line 382. “are said to” is very colloquial, so rephrase the sentence.

Answer: Desired changes have been incorporated.

Following all queries and suggestions, we have modified article. Kindly see changes and review at your earliest.

Regards,

Dr. Muhammad Farooq Hussain Munis

(CorrespondSing Author)

---

## [Decision Letter · Decision Letter 1]

22 Jan 2020

Resistance Associated Metabolite Profiling of Aspergillus Leaf Spot in Cotton through Non-targeted Metabolomics

PONE-D-19-26966R1

Dear Dr. Munis,

We are pleased to inform you that your manuscript has been judged scientifically suitable for publication and will be formally accepted for publication once it complies with all outstanding technical requirements.

With kind regards,

Kandasamy Ulaganathan

Academic Editor

PLOS ONE

Additional Editor Comments (optional):

Reviewers' comments:

Reviewer's Responses to Questions

**Comments to the Author**

1. If the authors have adequately addressed your comments raised in a previous round of review and you feel that this manuscript is now acceptable for publication, you may indicate that here to bypass the “Comments to the Author” section, enter your conflict of interest statement in the “Confidential to Editor” section, and submit your "Accept" recommendation.

Reviewer #1: All comments have been addressed

2. Is the manuscript technically sound, and do the data support the conclusions?

Reviewer #1: Yes

3. Has the statistical analysis been performed appropriately and rigorously? 

Reviewer #1: Yes

4. Have the authors made all data underlying the findings in their manuscript fully available?

Reviewer #1: Yes

5. Is the manuscript presented in an intelligible fashion and written in standard English?

Reviewer #1: Yes

6. Review Comments to the Author

Reviewer #1: In this version, the authors revised carefully according to the suggestions. The quality of this manuscript is significantly improved. Now, it meets the requirement of publication. Thus, I recommend it is accepted for publication.

7. PLOS authors have the option to publish the peer review history of their article (what does this mean?). If published, this will include your full peer review and any attached files.

Reviewer #1: Yes: Hongmei Cheng

---

## [Editor Report · Acceptance letter]

24 Jan 2020

PONE-D-19-26966R1 

Resistance Associated Metabolite Profiling of Aspergillus Leaf Spot in Cotton through Non-targeted Metabolomics 

Dear Dr. Munis:

I am pleased to inform you that your manuscript has been deemed suitable for publication in PLOS ONE. Congratulations! Your manuscript is now with our production department. 

With kind regards,

on behalf of

Dr. Kandasamy Ulaganathan 

Academic Editor

PLOS ONE